Integrated population modelling reveals potential drivers of demography from partially aligned data: a case study of snowy plover declines under human stressors

Zhao Qing 1 zhaoqin@missouri.edu
Heath-Acre Kristen 1 2
Collins Daniel 3
Conway Warren 2
Weegman Mitch D. 4
1 University of Missouri , Columbia, Missouri , United States
2 Texas Tech University , Lubbock, Texas , United States
3 US Fish & Wildlife Service , Albuquerque, New Mexico , United States
4 University of Saskatchewan , Saskatoon, Saskatchewan , Canada
Li Chenxi
Electronic publication date: 2021 Nov 15
Publication date: 2021
Volume: 9
Electronic Location ID: e12475
Received 2021 Aug 23; Accepted 2021 Oct 20
Copyright: © 2021 Zhao et al.
Copyright year: 2021
Copyright holder: Zhao et al.
License: This is an open access article distributed under the terms of the Creative Commons Attribution License, which permits unrestricted use, distribution, reproduction and adaptation in any medium and for any purpose provided that it is properly attributed. For attribution, the original author(s), title, publication source (PeerJ) and either DOI or URL of the article must be cited.
License URL: https://creativecommons.org/licenses/by/4.0/

Keywords: Climate change, Conservation, Data integration, Demography, Human stressor, Imbalanced sampling, Population monitoring, Wetland

Funding: U.S. Fish & Wildlife Service The research was funded by U.S. Fish & Wildlife Service. The funders had no role in study design, data collection and analysis, decision to publish, or preparation of the manuscript.

==============================
Knowledge of demography is essential for understanding wildlife population dynamics and developing appropriate conservation plans. However, population survey and demographic data (e.g., capture-recapture) are not always aligned in space and time, hindering our ability to robustly estimate population size and demographic processes. Integrated population models (IPMs) can provide inference for population dynamics with poorly aligned but jointly analysed population and demographic data. In this study, we used an IPM to analyse partially aligned population and demographic data of a migratory shorebird species, the snowy plover (Charadrius nivosus). Snowy plover populations have declined dramatically during the last two decades, yet the demographic mechanisms and environmental drivers of these declines remain poorly understood, hindering development of appropriate conservation strategies. We analysed 21 years (1998–2018) of partially aligned population survey, nest survey, and capture-recapture-resight data in three snowy plover populations (i.e., Texas, New Mexico, Oklahoma) in the Southern Great Plains of the US. By using IPMs we aimed to achieve better precision while evaluating the effects of wetland habitat and climatic factors (minimum temperature, wind speed) on snowy plover demography. Our IPM provided reasonable precision for productivity measures even with missing data, but population and survival estimates had greater uncertainty in years without corresponding data. Our model also uncovered the complex relationships between wetland habitat, climate, and demography with reasonable precision. Wetland habitat had positive effects on snowy plover productivity (i.e., clutch size and clutch fate), indicating the importance of protecting wetland habitat under climate change and other human stressors for the conservation of this species. We also found a positive effect of minimum temperature on snowy plover productivity, indicating potential benefits of warmth during night on their population. Based on our results, we suggest prioritizing population and capture-recapture surveys for understanding population dynamics and underlying demographic processes when data collection is limited by time and/or financial resources. Our modelling approach can be used to allocate limited conservation resources for evidence-based decision-making.

Introduction

The knowledge of demographic processes is fundamental to learning about natural populations (Turchin, 2003; Rockwood, 2015). Due to continuous impacts of global change such as climate change and agricultural development on biodiversity patterns (Thomas et al., 2004; Foley et al., 2005; Parmesan, 2006; Grimm et al., 2013) including population dynamics (Sæther, Sutherland & Engen, 2004; Reist et al., 2006; Simmonds & Isaac, 2007; Zhao et al., 2019), knowledge of demographic responses to environmental factors holds essential value in guiding conservation planning (Clark et al., 2001; Rushing et al., 2020).

It can be challenging to quantify population dynamics and underlying demographic processes with severely limited data, which are commonly encountered in population studies. Inferences about demographic processes (e.g., survival) often relies on data of marked animals (e.g., capture-recapture data; Pollock, 1991; Williams, Nichols & Conroy, 2002), which can be difficult to collect. Furthermore, demography and population survey data are not always spatially and temporally aligned. Approaches that link population survey and demographic data are particularly useful when data are relatively sparse because they can potentially provide a comprehensive understanding of population dynamics and underlying demographic processes.

Integrated population models (IPMs) jointly analyse multiple types of data such as population survey, capture-recapture, and productivity information (Besbeas et al., 2002; Brooks, King & Morgan, 2004; Schaub & Abadi, 2011). These models can provide more accurate and precise parameter estimates than models that analyse each data type separately (Abadi et al., 2010; Schaub & Abadi, 2011). Furthermore, IPMs can provide estimates of some parameters without direct data via sharing of information among data types (Besbeas et al., 2002; Tavecchia et al., 2009; Schaub, Jakober & Stauber, 2013; Zhao, Boomer & Royle, 2019). Consequently, IPMs are particularly useful when data are sparse and unaligned or partially aligned in space and time (Schaub et al., 2007; Davis et al., 2014; Saunders et al., 2019). IPMs have largely improved our understanding of natural animal populations (Schaub & Fletcher, 2015; Ahrestani et al., 2017; Weegman et al., 2017; Zhao et al., 2019) and allow for the development of more effective and efficient conservation practices (Arnold et al., 2018; Zipkin & Saunders, 2018; Zhao et al., 2020).

Many shorebird populations are sensitive to climate change (Van de Pol et al., 2010; Lehikoinen et al., 2013) because their key habitats (wetlands) are driven by weather and climatic factors (Sorenson et al., 1998; Sofaer et al., 2016; Zhao et al., 2016). Other types of human disturbance such as agricultural development may also lead to wetland habitat loss (Johnston, 2013; Burgin, Franklin & Hull, 2016; Donnelly et al., 2020). Consequently, shorebirds are threatened by multiple human stressors. For example, numerous North American shorebird populations have declined during the past half-century, likely due to degradation, fragmentation, and other kinds of human disturbance of wetlands (Howe, Geissler & Harrington, 1989; Bart et al., 2007; Rosenberg et al., 2019). However, warming temperatures resulting from climate change may drive shorebird population dynamics by influencing incubation behaviour and partitioning of incubation duties, particularly during cold periods (e.g., at night), because warmer temperatures allow birds to reserve more energy for reproduction or survival (Van de Pol et al., 2010; Saalfeld et al., 2012). Additionally, greater wind speed during breeding seasons may increase physiological stress during incubation and accelerate water evaporation, and thus negatively impact shorebird demography and incubation success (Høyvik Hilde et al., 2016).

The snowy plover (Charadrius nivosus) is a migratory shorebird species with breeding and wintering populations distributed along the Pacific Coast and Gulf Coast, as well as interior breeding populations in the Great Basin and Southern Great Plains (Page et al., 2009). Recent studies have estimated a severe decline of the interior breeding populations in the Southern Great Plains (Andres et al., 2012; Saalfeld et al., 2013a; Heath, 2019). Knowledge of the demographic causes and potential drivers of such a decline is essential for conservation planning of this species. However, knowledge gaps due to data limitations have hindered the development of effective conservation strategies.

In this study we used an IPM to analyse 21 years (i.e., 1998–2018) of partially aligned data for snowy plovers breeding within the Southern Great Plains. By using this modelling approach, we first aimed to achieve better precision of population and demographic estimates. We then evaluated the contributions of demographic processes to population growth. Lastly, we tested hypotheses regarding the drivers of productivity and survival, including wetland habitat, temperature, and wind speed. Based on our results, we provided recommendations for future population monitoring and conservation planning of snowy plover, and suggested prioritization of data collection schemes for conservation projects that often have limited resources.

Methods

Study area

Our study is located in the ecological region of the Southern Great Plains in Texas, New Mexico and Oklahoma (Fig. 1), which encompasses semi-arid short and mixed grass prairie (Assal, Melcher & Carr, 2015). More specifically, we studied three breeding populations in Texas, New Mexico and Oklahoma, respectively. Study sites included three privately owned saline lakes (i.e., A, B and C) and Muleshoe National Wildlife Refuge (NWR) in Texas, Bitter Lake NWR in New Mexico, and Salt Plains NWR in Oklahoma. Details about the study sites can be found in Heath (2019).

Figure 1 Study area.

The position of the Great Plains in the contiguous US (inner panel), and the positions of our study sites in the Southern Great Plains in Texas (Muleshoe National Wildlife Refuge (NWR) and lakes A, B and C), New Mexico (Bitter Lake NWR), and Oklahoma (Salt Plains NWR).

Data collection

Population survey

For the Texas population, surveys were conducted weekly May through July at lakes A, B and C in 1998–2000, 2008–2010, and 2017–2018 (Fig. 2), along transects that covered 3.2–3.5 km sections of shoreline (Heath, 2019). Multiple observers who have conducted the surveys were well trained to have consistent survey abilities. Survey areas were consistent within each year and among years. Surveys began at approximately 08:00 and lasted 1–2 h in days without abnormally high winds (i.e., wind speed > 50 mph) or rain. For the New Mexico population, surveys were conducted biweekly and otherwise under the same protocol of the Texas population, in each year from 1999 through 2018.

Figure 2 Data availability.

The years in which each type of data (population survey, distance sampling, nest survey, and capture-recapture-resight) are available for each population (Texas, New Mexico, Oklahoma).

For the Oklahoma population, annual surveys were conducted on a single day in early May from 2013 to 2017 at Salt Plains NWR. The entire salt flat area of Salt Plains NWR was divided into a total of 668 grids that were 300 m × 300 m, among which 100 were randomly selected for surveys. Biologists and volunteers were paired, and 10–12 grid cells were assigned to each pair to survey. Surveys were conducted along 300 m transects while birds within 75 m distance from the transects were counted. In addition to annual surveys, distance sampling was conducted at Salt Plains NWR during May–July in 2017 and 2018. The region was divided into three sub-regions (i.e., north, middle, south). The same grids of the annual survey were used, among which nine (3 for north, 2 for middle, 4 for south) were randomly selected. Surveys were again conducted along 300 m transects, but in addition to counting birds with 75 m distance, the linear distance between the observed birds and the transects was also recorded.

More details about the population surveys can be found in Saalfeld et al. (2013a) and Heath-Acre et al. (2021).

Nest survey

We surveyed snowy plover nests at least once per week during the breeding season. Nests were located by searching suitable habitat and observing adults incubating nests or flushing from or returning to nests. The search effort was relatively consistent among study sites and years (Conway, Smith & Ray, 2005). Once a nest was located, clutch size (i.e., the number of eggs) and ultimately clutch fate (success or failure) were determined and recorded. Nest surveys were conducted in 1999–2000, 2008–2009, and 2017–2018 at lakes A, B and C, and in 1999–2000 and 2008–2009 at Muleshoe NWR for the Texas population, in 2017–2018 at Bitter Lake NWR for the New Mexico population, and in 2017–2018 at the Salt Plains NWR for the Oklahoma population.

Capture-recapture-resight

Adult snowy plovers were captured at feeding locations using mist nets and on nests using nest traps (Conway & Smith, 2000). Juveniles were captured within 24 h of hatching when they were still close to the nest and had not moved into foraging grounds, either by hand in nests or with adult(s) after hatching. All captured individuals were banded with a uniquely numbered U.S. Geological Survey aluminium band and a unique combination of colour bands. Blood samples were collected during captures to identify sex (Saalfeld et al., 2013a). The identities of banded birds were recorded during subsequent captures or population surveys (see above), yielding recapture and resighting information. The capture-recapture-resight surveys were conducted in 1999–2000, 2008–2009, 2013–2014, and 2016–2018 at lakes A, B and C and Muleshoe NWR for the Texas population, in 2013–2014 and 2017–2018 at the Bitter Lake NWR for the New Mexico population, and in 2013–2018 at the Salt Plains NWR for the Oklahoma population.

Environmental data

We considered Palmer drought severity index, minimum temperature, and wind speed as potential drivers of snowy plover productivity and survival. Palmer drought severity index is a measurement of the amount of surface water based on recent precipitation and temperature (Palmer, 1965). As Palmer drought severity index tends to be positively correlated with precipitation and negatively correlated with maximum temperature (Fig. S1), it can be used to represent wetland habitat availability (i.e., a higher Palmer drought severity index value indicates greater wetland habitat availability) that influences shorebird demography (Todhunter, 1995; Dinsmore, 2008). High minimum temperature represented warmth during night, which may influence snowy plover demography through energy reserves. We considered wind speed because greater wind speed may increase physiological stress during incubation and thus negatively influence snowy plover survival and productivity. We initially also considered actual evapotranspiration, precipitation, and maximum temperature, but these variables were highly correlated with Palmer drought severity index and/or minimum temperature (Fig. S1), and thus were not included in the model. We calculated the mean values of the above-mentioned covariates during the breeding season (i.e., May–July) for each population and year (Fig. S2) and included them in our IPM.

Modelling approach

We used an IPM to explain snowy plover population dynamics as a consequence of the spatiotemporal variation in productivity and survival. Our IPM included three sub-models: a population sub-model, a productivity sub-model, and a survival sub-model, which utilized population survey, nest survey, and capture-recapture-resight data, respectively. We describe each sub-model and then the overall model below.

Population sub-model

We assumed that population size in region i in the first year, denoted Ni,1, followed a log-Normal distribution such that log(Ni,1)∼Normal(μi[0],0.1), in which μi[0] was selected based on the population survey data of the corresponding region, and a small standard deviation of 0.1 was used. From the second year (i.e., t≥2), population size was assumed to follow a log-Normal distribution such that

(1) log(Ni,t)=log(Ni,t−1×0.5×ϕi,t−1[AM]+Ni,t−1×0.5×ϕi,t−1[AF]+Ni,t−1×0.5×γi,t−1×πi,t−1×0.5×ϕi,t−1[JM]+Ni,t−1×0.5×γi,t−1×πi,t−1×0.5×ϕi,t−1[JF])+εi,t[N],

in which ϕi,t−1[AM], ϕi,t−1[AF], ϕi,t−1[JM], and ϕi,t−1[JF] were apparent survival of adult males, adult females, juvenile males, and juvenile females, respectively, γi,t−1 was average clutch size, πi,t−1 was clutch fate, and εi,t[N] were process errors that followed a Normal distribution of mean 0 and standard deviation σ[N]. The process errors εi,t[N] represented the effects of demographic stochasticity. Note that we assumed that all juveniles breed at one year old (Warriner et al., 1986). We also assumed that both adult sex ratio and clutch sex ratio were 1:1 (Saalfeld et al., 2013b). We did not consider partial nest mortality because it was rare in our study populations (Conway personal communication). We ignored immigration because we did not observe movement among our study populations during our study.

We then linked the observed population survey data with the true but latent population size by specifying an observation model that was specific to each data set that we fused in our IPM. For the Texas and New Mexico populations, we assumed that population counts followed Poisson log-Normal distributions such that yi,t,l,k∼Poisson(Ni,t×exp[εi,t,l,k[y]]×p[COUNT]), in which yi,t,l,k was the population count of population i in year t at lake l and on date k, εi,t,l,k[y] were lake-and date-specific variation in local abundance that followed a Normal distribution of mean 0 and standard deviation σ[y]. The detection probability p[COUNT] for these surveys was assumed to be a constant that equalled the detection probability at the median distance ( d¯ = 30 m) in distance sampling (see below) such that p[COUNT]=exp(−1×ξ×d¯).

For the Oklahoma population, we assumed that the grid-level (indexed by j) counts followed a Poisson log-Normal distribution such that yi,t,j∼Poisson(Ni,t668×exp[εi,t,j[y]]×p[COUNT]) for the annual surveys from 2013 to 2017, in which the total population size was divided by the total number of grids (i.e., 668, see above), εi,t,j[y] represented the variation in local abundance among grid j, and p[COUNT] again was the constant detection probability for these annual surveys. We also assumed yi,t,j∼Poisson(Ni,t668×exp[εi,t,j[y]]×pi,t,j[DIST]) for the distance sampling in 2017 and 2018. Here we had pi,t,j[DIST]=exp(−1×ξ×di,t,j), in which ξ was a decay parameter representing the assumption that detection probability would decrease when the linear distance between the birds and the transect, denoted di,t,j, increased (Royle, Dawson & Bates, 2004; Schmidt & Rattenbury, 2018).

Productivity sub-model

We assumed that clutch size and fate were functions of Palmer drought severity index, minimum temperature and wind speed. More specifically, we linked clutch size with these covariates using a multinomial logistic regression. We considered the probability of a clutch size n (denoted ωi,t[n]) for n = 1, 2, and 3. Average clutch size was then calculated as γi,t=1×ωi,t[1]+2×ωi,t[2]+3×ωi,t[3]. We then had ωi,t[n]=θi,t[n]∑n=13θi,t[n], in which θi,t[1] = 1, and θi,t[2] and θi,t[3] were expressed as functions of the above-mentioned covariates such that

(2) log(θi,t[2])=α[2]+β1[2]×PDSIi,t+β2[2]×MINTi,t+β3[2]×WINDi,t+εi,t[2],

and

(3) log(θi,t[3])=α[3]+β1[3]×PDSIi,t+β2[3]×MINTi,t+β3[3]×WINDi,t+εi,t[3],

in which PDSIi,t was the Palmer drought severity index for region i and year t, MINTi,t was minimum temperature, WINDi,t was wind speed, and εi,t[2] and εi,t[3] were process errors that followed Normal distributions of mean 0 and standard deviations of σ[2] and σ[3], respectively. Note that we used intercept and slope parameters that were the same across the regions, which represented the assumption the demography-environment relationships were the same across regions. We also considered regressions with region-specific intercept and slope parameters such that log(θi,t[2])=αi[2]+β1,i[2]×PDSIi,t+β2,i[2]×MINTi,t+β3,i[2]×WINDi,t+εi,t[2] and log(θi,t[3])=αi[3]+β1,i[3]×PDSIi,t+β2,i[3]×MINTi,t+β3,i[3]×WINDi,t+εi,t[3] to allow for region-specific demography-environment relationships.

We linked clutch fate with the same covariates using a logistic regression such that

(4) logit(μi,t)=α[f]+β1[f]×PDSIi,t+β2[f]×MINTi,t+β3[f]×WINDi,t+εi,t[f],

in which process errors εi,t[f] followed a Normal distribution with mean 0 and standard deviations of σ[f]. As in the regressions of clutch size, we also formed a regression with region-specific intercept and slope parameters for clutch fate.

Survival sub-model

We linked apparent survival with the same covariates mentioned above using a logistic regression such that

(5) logit(ϕi,t−1[C])=α[C]+β1[C]×PDSIi,t+β2[C]×MINTi,t+β3[C]×WINDi,t+εi,t[C],

in which ϕi,t−1[C] was the apparent survival of cohort C (i.e., adult male, adult female, juvenile male, juvenile female) in region i and year t, and process errors εi,t[C] followed a Normal distribution with mean 0 and standard deviations of σ[C]. We again considered a regression with region-specific intercept and slope parameters for apparent survival.

We also estimated the probability of recapture ( p[REC]) and resighting ( p[RES]). The likelihood of the individual encounter history data was then calculated using ϕi,t−1[C], p[REC], and p[RES] values.

Model implementation

We implemented the IPM in a hierarchical Bayesian framework with posterior distributions obtained by Markov chain Monte Carlo (MCMC) computing in the software JAGS (Plummer, 2003), which was called from R (R Development Core Team, 2013) through the package “jagsUI” (Kellner, 2015). We used vague priors Normal (0, 100) for any intercept and slope parameters, Gamma (0.01, 0.01) for any precision parameters and the decay parameter in distance sampling, and Uniform (0, 1) for capture and resighting probability parameters. We used four chains with 300,000 iterations including 50,000 burn-in and thinned by 100, yielding 10,000 posterior samples for each parameter. We checked the convergence of the MCMC computing using R-hat statistics and Gelman-Rubin diagnostics (Brooks & Gelman, 1998). The R-hat statistics for each parameter were ≤1.02, and the chains were well mixed.

Post-modelling analysis

We conducted posterior predictive checks to examine model fit (Gelman et al., 2004). We used Chi-square statistics as the discrepancy measures for the population survey and distance sampling data (Gelman, Meng & Stern, 1996). We used the Freeman-Tukey statistic as the discrepancy measures for the clutch size, clutch fate, and capture-recapture-resighting data (Brooks et al., 2000). We calculated these discrepancy measures for both the actual data and simulated data using the full posterior samples. We then calculated posterior predictive P-values as the proportion of discrepancy measures that were larger for the simulated data than for the actual data. A posterior predictive P-value that is close to 0.5 indicates a good model fit.

We conducted a hierarchical partitioning analysis (Mac Nally, 1996) to understand the relative contributions of demographic parameters in describing population growth rates (Zhao, Boomer & Royle, 2019). Hierarchical partitioning is based on multiple linear regressions, in which population growth (i.e., Ni,tNi,t−1) was the response variable and demographic parameters were the predictors. This approach considers all possible models, each of which corresponds to a given combination of predictors. For each model, the joint contribution of the predictors is calculated. With such information, hierarchical partitioning allowed us to calculate the relative independent contributions of each demographic parameter on population growth. We used the full posterior samples to conduct these analyses to account for uncertainty in parameter estimates.

We calculated and reported standardized effect sizes (SES) for slope parameters representing the effects of environmental covariates on demographic parameters. The standardized effect size was defined as the absolute ratio of posterior mean to the width of 80% Credible Interval. We considered the effect of a covariate “strong” when SES was >0.6, “moderate” when SES was 0.3–0.6, and “little to no effect” when SES was <0.3.

Results

Population and demographic estimates

Posterior predictive check indicated good model fit for all sub-models (population survey and distance sampling: posterior predictive P-value = 0.49; clutch size: P-value = 0.42; clutch fate: P-value = 0.58; capture-recapture-resighting: P-value = 0.62).

Our results revealed population declines in the Texas and New Mexico populations, and potentially the Oklahoma population (Fig. 3). Note that the Oklahoma population had a much larger population size (e.g., mean 1,815.0, 80% Credible Interval C.I. [1,590.1–2,081.2] in 2017) than the Texas (mean 83.7, 80% C.I. [75.2–93.4] in 2017) and New Mexico populations (mean 23.9, 80% C.I. [20.8–27.4] in 2017). However, the trend of the Oklahoma population was less clear due to the lack of population survey data in the early years. For the population surveys, detection probability averaged 0.999 (80% C.I. [0.983–1.000]), indicating nearly perfect detection. For the distance sampling, the decay parameter averaged 0.020 (80% C.I. [0.000–0.047]), indicating weak decrease in detection probability when distance increased.

Figure 3 Population estimates.

IPM estimated population trend (yellow line) and corresponding uncertainty gradient (purple strip) as well as population count data (red points) in Texas, New Mexico, and Oklahoma and distance sampling data (blue points) in Oklahoma.

Average clutch size (Texas: mean 2.63, 80% C.I. [2.03–2.97]; New Mexico: mean 2.67, 80% C.I. [2.03–2.97]; Oklahoma: mean 2.62, 80% C.I. [2.00–2.98]) and clutch fate (Texas: mean 0.37, 80% C.I. [0.09–0.84]; New Mexico: mean 0.40, 80% C.I. [0.08–0.83]; Oklahoma: mean 0.40, 80% C.I. [0.05–0.92]) were similar among populations (Fig. 4). Apparent survival of each cohort was also similar among populations, but adult female (Texas: mean 0.75, 80% C.I. [0.19–0.99]; New Mexico: mean 0.78, 80% C.I. [0.18–0.99]; Oklahoma: mean 0.75, 80% C.I. [0.13–0.99]) and adult male (Texas: mean 0.78, 80% C.I. [0.16–0.99]; New Mexico: mean 0.79, 80% C.I. [0.15–0.99]; Oklahoma: mean 0.74, 80% C.I. [0.16–0.99]) has on average greater survival than juvenile female (Texas: mean 0.17, 80% C.I. [0.00–0.88]; New Mexico: mean 0.14, 80% C.I. [0.00–0.89]; Oklahoma: mean 0.10, 80% C.I. [0.00–0.89]) and juvenile male (Texas: mean 0.07, 80% C.I. [0.00–0.74]; New Mexico: mean 0.09, 80% C.I. [0.00–0.69]; Oklahoma: mean 0.06, 80% C.I. [0.00–0.66]; Fig. 5) in all study populations, with high uncertainty.

Figure 4 Productivity estimates.

IPM estimated average clutch size and clutch fate (yellow line) and corresponding uncertainty gradient (purple strip) in Texas, New Mexico, and Oklahoma. Sample size (i.e., the number of nests surveyed; blue line) is also shown.

Figure 5 Survival estimates.

IPM estimated apparent survival of adult female, adult male, juvenile female and juvenile male (yellow line) and corresponding uncertainty gradient (purple strip) in Texas, New Mexico, and Oklahoma. Sample size (i.e., the number of banded birds; blue line) is also shown.

Demographic contributions on population growth

Demographic parameters considered in our IPM had similar but uncertain contributions to population growth rates (clutch size: mean 8.8%, 80% C.I. [2.0%–31.5%], clutch fate: mean 10.5%, 80% C.I. [2.5%–37.8%], adult female survival: mean 12.2%, 80% C.I. [2.2%–45.8%], adult male survival: mean 12.1%, 80% C.I. [2.2%–45.6%], juvenile female survival: mean 10.9%, 80% C.I. [2.3%–40.3%], juvenile male survival: mean 11.4%, 80% C.I. [1.9%–41.5%]; Fig. 6).

Figure 6 Demographic contributions.

The relative independent contribution of average clutch size, clutch fate, and apparent survival of adult female, adult male, juvenile female, and juvenile male on snowy plover population growth in Texas, New Mexico, and Oklahoma. The violin plot shows the entire posterior distribution, while the embedded boxplot shows the median (white dot), 50% Credible Interval (thick line), and 95% Credible Interval (thin line).

Drivers of demography

The results from the IPM with region-specific intercept and slope parameters showed little regional variation in demography-environment relationships (Figs. S3 & S4). Therefore, here we report the results of the model with universal intercept and slope parameters.

Palmer drought severity index (recall that a higher Palmer drought severity index indicates higher wetland habitat availability) had a moderate positive effect on the probability of a clutch size of 2 (mean 0.71, 80% C.I. [−0.29 to 1.91], SES = 0.32), a strong positive effect on the probability of a clutch size of 3 (mean 1.59, 80% C.I. [0.56–2.91], SES = 0.68), and a strong positive effect on clutch fate (mean 1.77, 80% C.I. [1.27–2.33], SES = 1.67; Figs. 7 & 8). Minimum temperature had little to no effect on the probability of a clutch size of 2 (mean 0.23, 80% C.I. [−0.26 to 0.78], SES = 0.22), a strong positive effect on the probability of a clutch size of 3 (mean 0.82, 80% C.I. [0.34–1.37], SES = 0.80), and a strong positive effect on clutch fate (mean 0.99, 80% C.I. [0.70–1.28], SES = 1.69; Figs. 7 & 8). Wind speed had a moderate positive effect on clutch fate (mean 0.38, 80% C.I. [0.06–0.80], SES = 0.52), but had little to no effect on the probability of a clutch size of 2 (mean 0.09, 80% C.I. [−0.56 to 0.80], SES = 0.06) or 3 (mean 0.12, 80% C.I. [−0.51 to 0.84, SES = 0.09; Fig. 7).

Figure 7 Drivers of productivity.

Violin plots showing the posterior distributions of the slope parameters that represent the effect of Palmer drought severity index (PDSI), minimum temperature (min temp), and wind speed (wind) on productivity measures of snowy plover including the probability of a clutch size of 2 (C2) or 3 (C3), and clutch fate (FT). The violin plot shows the entire posterior distribution, while the embedded boxplot shows the median (white dot), 50% Credible Interval (thick line), and 95% Credible Interval (thin line).

Figure 8 Response curves for productivity.

Response curves showing the effects of Palmer drought severity index (PDSI) and minimum temperature (min temp) on average clutch size and clutch fate. The predicted relationships (yellow line) and corresponding uncertainty gradient (purple strip) for the estimated demographic parameters (blue point/line) are shown.

Palmer drought severity index had little to no effect on adult female survival (mean 0.45, 80% C.I. [−0.45 to 1.28], SES = 0.26) and adult male survival (mean 0.21, 80% C.I. [−0.99 to 1.06], SES = 0.10), but had moderate positive effects on juvenile female survival (mean 1.12, 80% C.I. [0.13–3.22], SES = 0.36) and juvenile male survival (mean 0.70, 80% C.I. [−0.30 to 1.84], SES = 0.33). Minimum temperature (adult female: mean 0.00, 80% C.I. [−0.66 to 0.75], SES = 0.00; adult male: mean −0.35, 80% C.I. [−1.24 to 0.35], SES = 0.22; juvenile female: mean 0.18, 80% C.I. [−0.67 to 1.88], SES = 0.07; juvenile male: mean −0.39, 80% C.I. [−1.84 to 0.49], SES = 0.17) and wind speed (adult female: mean −0.06, 80% C.I. [−0.75 to 0.57], SES = 0.05; adult male: mean −0.07, 80% C.I. [−0.78 to 0.55, SES = 0.05; juvenile female: mean 0.04, 80% C.I. [−0.65 to 0.77], SES = 0.03; juvenile male: mean −0.02, 80% C.I. [−0.71 to 0.66], SES = 0.02) had little to no effect on survival (Fig. 9).

Figure 9 Drivers of survival.

Violin plots showing the posterior distributions of the slope parameters that represent the effect of Palmer drought severity index (PDSI), minimum temperature (min temp), and wind speed (wind) on the apparent survival of adult female (AF), adult male (AM), juvenile female (JF), and juvenile male (JM) snowy plover. The violin plot shows the entire posterior distribution, while the embedded boxplot shows the median (white dot), 50% Credible Interval (thick line), and 95% Credible Interval (thin line).

Discussion

Our IPM provided reasonable precision for productivity estimates and uncovered the complex relationships between wetland habitat conditions, climate, and demography with partially aligned data. Our results showed that wetland habitat may have positively impacted productivity of snowy plover, indicating the importance of protecting wetland habitat for the conservation of this migratory shorebird that breeds in a semi-arid environment. Our results also showed that minimum temperature may have positively influenced productivity. We recommend to prioritize data collection on population and capture-recapture surveys for understanding population dynamics and underlying demographic processes when data collection is limited by time and/or financial resource.

Analysis of partially aligned data

IPMs have been increasingly used in understanding population dynamics and underlying demographic processes due to their capability of estimating parameters with unbalanced data (Schaub et al., 2007; Davis et al., 2014; Saunders et al., 2019) or even without specific data (Besbeas et al., 2002; Zhao, Boomer & Royle, 2019), at no substantial cost to bias or precision of parameter estimates (Weegman et al., 2020). It is straightforward to analyse partially aligned data with IPMs under the hierarchical Bayesian framework as the modelling approach allows for missing values in most, if not all, parts of the model.

Furthermore, it is feasible to incorporate information-borrowing mechanisms among regions in IPMs that is particularly important when data are sparse and/or partially aligned. Borrowing information can be achieved by using universal or region-specific slope parameters; while the former provides a stronger information-borrowing mechanism, the latter is flexible to reflect region-specific demography-environment relationships. Previous work revealed that the responses of populations to environmental conditions may be region-specific (Forchhammer et al., 1998; Williams, Ives & Applegate, 2003; Grøtan et al., 2009). While these studies often focus on large spatial ranges that cover multiple ecological regions, Zhao, Boomer & Royle (2019) found that demography-environment relationships tend to be similar within, but not among, ecological regions. Because our study area lies in one ecological region (i.e., the Southern Great Plains), it is not surprising to find that our three populations respond similarly to the environment. Therefore, we used a model with universal demography-environment relationships, which allows us to achieve relatively reasonable precision, particularly for productivity estimates, for all three populations. We encourage practitioners to consider information-borrowing approaches among populations when data are only partly aligned, such as in our study.

For our case study, uncertainty of population and survival estimates was still high in some years due to the lack of corresponding data, even with an advanced IPM and information-borrowing. For example, Oklahoma population estimates from 1998 to 2012 had relatively high uncertainty due to lack of population survey data during this period. These results remind us that monitoring programs are still extremely important for gaining knowledge about wildlife populations, even with advantages from recent modelling techniques.

Environmental impacts

Our study revealed potential environmental drivers of demography in three declining snowy plover populations in the Southern Great Plains (Andres et al., 2012; Saalfeld et al., 2013a; Heath, 2019). Our study showed that wetland habitat had a strong positive effect on snowy plover productivity measures (i.e., clutch size and clutch fate) and a moderate positive effect on juvenile survival. Thus, the declines in snowy plover productivity and population size can be attributed, in part, to insufficient wetland habitat. Like other shorebirds, snowy plover populations rely on wetland habitat (Conway, Smith & Ray, 2005), where degradation or loss in wetland habitat may decrease their productivity or even survival (Saalfeld et al., 2011; Saalfeld et al., 2013a). Wetland habitat loss could be driven by climate change (e.g., Sorenson et al., 1998; Sofaer et al., 2016) as well as other human stressors (Johnston, 2013; Burgin, Franklin & Hull, 2016; Donnelly et al., 2019; Donnelly et al., 2020). For example, the decline of the snowy plover population at Bitter Lake NWR may be driven by the degradation of ground water sources in the Pecos River ecosystem related to agricultural development (Heath, 2019).

Our study also reveals a positive effect of minimum temperature on snowy plover clutch size and fate. Increases in temperature are normally considered to lead to drier habitats and thus negatively impact shorebird populations. In our study area, maximum temperature is negatively correlated with wetland habitat, represented by Palmer drought severity index (Fig. S1). Even though we could not evaluate the effect of maximum temperature on snowy plover demography due to this correlation, the positive effect of wetland habitat may indicate a negative impact of high temperature during day time on snowy plover demography. High temperatures not only could lead to increased evapotranspiration and drought but also may create a thermally stressful environment for nesting snowy plovers that necessitates incubating parents cooling eggs during daylight hours (Saalfeld et al., 2012). High minimum temperature, on the other hand, represents a relatively warm condition during night, which may benefit these birds (Van de Pol et al., 2010; Saalfeld et al., 2012). Further studies that are able to disentangle the multifaceted effects of climatic conditions on shorebird demography and behaviours are essential for a comprehensive understanding of the impacts of anticipated change on their populations, associated with climate change. Despite that we predicted a negative effect of wind speed on snowy plover demography (Høyvik Hilde et al., 2016), we found that wind speed had a moderate positive effect on clutch fate but had no effect on other demographic parameters. However, the relatively high uncertainty of the effect of wind speed indicated that further investigation is needed.

Interestingly, our results showed that productivity and survival of all cohorts had similar contributions on snowy plover population growth. Several studies showed that productivity tends to vary more and also contribute more to population growth (Alisauskas et al., 2004; Cooch, Rockwell & Brault, 2001; Taylor et al., 2012), although these studies focused on larger birds. Survival might play a more important role than productivity in smaller birds such as snowy plovers (current study) and insectivores (Schaub, von Hirschheydt & Grüebler, 2015). The relatively high uncertainty of population and survival estimates in some years, however, may have masked the differential contributions of demography on population growth, warranting further investigation.

Conservation implications

Conservation programs often have limited financial resources and practitioners are challenged to balance monitoring and conservation priorities. Demographic data (e.g., capture-recapture) are more expensive to collect than count data of unmarked populations yet are crucial for understanding demographic foundations of population dynamics. Our study showed that reasonable precision of productivity estimates could be achieved even though nest survey data were available for only short periods. Previous researchers showed that productivity/recruitment could be estimated without direct data using IPMs (e.g., Besbeas et al., 2002; Zhao, Boomer & Royle, 2019; Weegman et al., 2020). Survival estimates in Texas also had an overall reasonable precision despite the gaps in capture-recapture-resight data. Survival estimates in Oklahoma, however, had relatively high uncertainty during the first 15 years of our study period due to the lack of capture-recapture-resight data. Population estimates had relatively reasonable precision only for years with population survey data. In particular, the Oklahoma population was much larger than the other two populations and thus seems particularly important for the conservation of this species, which corroborates previous work (Heath, 2019). Yet the estimates of the Oklahoma population had high uncertainty during the early years due to the lack of data, which largely hindered our ability to identify a long-term trend for this population. Overall, it seems to be reasonable to prioritize limited time and financial resource on population and capture-recapture surveys for monitoring population status and developing appropriate conservation strategies for this species. On the other hand, it seems to be acceptable, although not preferable, to only collect nest survey data in some years.

Despite the imbalanced data availability, our IPM provided an understanding about the environmental drivers of snowy plover in the Southern Great Plains. Our study revealed that snowy plover demography and thus population dynamics are driven by wetland habitat conditions, indicating the importance of wetland habitat conservation under climate change and other human stressors such as groundwater mining and agricultural development (Conway, Smith & Ray, 2005; Heath, 2019). Our study also showed that future warming may potentially benefit snowy plover populations, at least in the short term (i.e., acknowledging that beyond a certain point, increased temperatures will negatively influence snowy plover productivity; Saalfeld et al., 2013a). Understanding the multifaceted effects of climate on animal demography is key for accurate forecasts of population responses, and thus appropriate conservation planning under climate change (Clark et al., 2001; Petchey et al., 2015).

Taken together, IPMs lay a foundation of allocating limited conservation resources for evidence-based decision-making. The benefits of IPMs are not limited to our study species, as more studies should consider balancing allocation of conservation resources among different types of data.

Supplemental Information

Supplemental Information 1 Population survey data in Texas.

year: the year in which the survey was conducted

lake: the lake where the survey was conducted

date: Julian date when the survey was conducted

count: count data

Click here for additional data file.

Supplemental Information 2 Population survey data in New Mexico.

year: the year in which the survey was conducted

date: Julian date when the survey was conducted

count: count data

Click here for additional data file.

Supplemental Information 3 Population survey data in Oklahoma.

year: the year in which the survey was conducted

grid: the grid cell in which the survey was conducted

count: count data

Click here for additional data file.

Supplemental Information 4 Distance sampling data in Oklahoma.

year: the year in which the survey was conducted

grid: the grid cell in which the survey was conducted

count: count data

distance: the distance between the observer and the counted birds

Click here for additional data file.

Supplemental Information 5 Encounter history data.

id: individual identity

cohort: age-sex class at the first capture

region: region of the survey, Texas, New Mexico, or Oklahoma

f: year of the first encounter

year_1998 ~ year_2018: encounter history from 1998 through 2018

Click here for additional data file.

Supplemental Information 6 Nest survey data.

year: the year in which the survey was conducted

region: region of the survey, Texas, New Mexico, or Oklahoma

clutch: clutch size, 1, 2, or 3

fate: clutch fate

Click here for additional data file.

Supplemental Information 7 Covariates data in Texas.

year: year

palmer: Palmer drought severity index

aet: actual evapotranspiration

prcp: precipitation

tmax: maximum temperature

tmin: minimum temperature

wind: wind speed

Click here for additional data file.

Supplemental Information 8 Covariates data in New Mexico.

year: year

palmer: Palmer drought severity index

aet: actual evapotranspiration

prcp: precipitation

tmax: maximum temperature

tmin: minimum temperature

wind: wind speed

Click here for additional data file.

Supplemental Information 9 Covariates data in Oklahoma.

year: year

palmer: Palmer drought severity index

aet: actual evapotranspiration

prcp: precipitation

tmax: maximum temperature

tmin: minimum temperature

wind: wind speed

Click here for additional data file.

Supplemental Information 10 Readme file for datasets.

Click here for additional data file.

Supplemental Information 11 Code of the model.

Click here for additional data file.

Supplemental Information 12 Correlations between covariates.

Scatter plots (upper triangle) and correlation coefficients (lower triangle) between each pair of the six environmental covariates that we originally considered, including Palmer drought severity index (PDSI), actual evapotranspiration (evap), precipitation (prec), maximum temperature (max temp), minimum temperature (min temp), and wind speed (wind).

Click here for additional data file.

Supplemental Information 13 Time series of covariates.

Time series of Palmer drought severity index (PDSI), minimum temperature (min temp), and wind speed (wind) in Texas, New Mexico, and Oklahoma for the study period.

Click here for additional data file.

Supplemental Information 14 Region-specific productivity-environment relationships.

Violin plots showing the region-specific effects of Palmer drought severity index (PDSI), minimum temperature (min temp), and wind speed (wind) on the probability of a clutch size of 2 or 3 and clutch fate in Texas (TX), New Mexico (NM), and Oklahoma (OK). The violin plot shows the entire posterior distribution, while the embedded boxplot shows the median (white dot), 50% Credible Interval (thick line), and 95% Credible Interval (thin line).

Click here for additional data file.

Supplemental Information 15 Region-specific survival-environment relationships.

Violin plots showing the region-specific effects of Palmer drought severity index (PDSI), minimum temperature (min temp), and wind speed (wind) on the apparent survival of adult female, adult male, juvenile female, and juvenile male in Texas (TX), New Mexico (NM), and Oklahoma (OK). The violin plot shows the entire posterior distribution, while the embedded boxplot shows the median (white dot), 50% Credible Interval (thick line), and 95% Credible Interval (thin line).

Click here for additional data file.

We thank Sarah Saalfeld, Hannah Ashbaugh, Laura Duffie and all field assistants who have contributed in data collection. M. Frederiksen, M. Paquet and M. Kery provided valuable comments on the manuscript.

Additional Information and Declarations

Competing Interests

Author Contributions

Data Availability

The authors declare that they have no competing interests.

Qing Zhao analyzed the data, prepared figures and/or tables, authored or reviewed drafts of the paper, and approved the final draft.

Kristen Heath-Acre conceived and designed the experiments, performed the experiments, authored or reviewed drafts of the paper, and approved the final draft.

Daniel Collins conceived and designed the experiments, authored or reviewed drafts of the paper, and approved the final draft.

Warren Conway conceived and designed the experiments, authored or reviewed drafts of the paper, and approved the final draft.

Mitch D. Weegman conceived and designed the experiments, authored or reviewed drafts of the paper, and approved the final draft.

The following information was supplied regarding data availability:

The raw data of population survey, capture-resight, and nest survey and our code to analyze these data are available in the Supplemental Files.

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
