# Peer review of "Integrated population modelling reveals potential drivers of demography from partially aligned data: a case study of snowy plover declines under human stressors"

_PeerJ, doi:10.7717/peerj.12475_

## Round 0.1 · original submission · Major Revisions

Dear Dr. Zhao,

Thank you for your submission to PeerJ.

It is my opinion as the Academic Editor for your article - Integrated population modelling improves the utility of partially aligned data - that it requires a number of Major Revisions.

Reviewers' comments on your work have now been received. The manuscript has been assessed by three reviewers. Reviews indicated that the populations modeled are not clearly defined, some assumptions of the model are not explicitly mentioned, and the way detection probability is handled in the population surveys is inconsistent. I agree with this evaluation and I would, therefore, request for the manuscript to be revised accordingly.

My suggested changes and reviewer comments are shown below and on your article 'Overview' screen. In addition, one of the reviewers has attached an annotated manuscript to this review.

Please address these changes and resubmit. Although not a hard deadline please try to submit your revision within the next 35 days.

With kind regards,
Chenxi Li
Academic Editor, PeerJ

·

Basic reporting

No comment.

Experimental design

The research question is relevant and well stated. As the authors argue, an Integrated Population Model (IPM) is a an appropriate tool for addressing the question (identifying mechanisms and drivers of population decline), not least because of the somewhat fragmentary nature of the available data. However, some aspects of the methods are not described in adequate detail, and some analytical choices can be questioned. Specifically:

1. The populations modeled are not clearly defined. It is not clear whether the study areas can be considered self-contained populations, or represent a sample of many other sites that also hold breeding snowy plovers. In either case, dispersal in to and out of the study sites is surely also possible, although this aspect is not included in the IPM. I think that a better description of the study sites, their importance for the species, and the arguments for including these specific sites is needed.
2. Some assumptions of the model are not explicitly mentioned. For instance, it appears from the very simple population model (l. 188-189) that all birds are assumed to breed when one year old. This assumption should be clearly stated and arguments provided. Generally, a bit more information about the study species and its life cycle should be provided. For instance, it is not necessarily clear to the reader that plover chicks are nidifugous, and that nest success thus equates hatching success, whereas post-hatching mortality must be estimated with capture-recapture methods.
3. The way detection probability is handled in the population surveys is inconsistent. In Texas and New Mexico, it appears that counts are taken as unbiased estimates of population size, i.e. detection probability is assumed to be 1.There is no information on what is actually counted in these surveys, e.g. whether all birds seen are included, regardless of distance to the observer? This point interacts with point 1 - defining the study area and population more clearly would be helpful. In this context, it is strange that some counts are up to ~10 times as high as the estimated population size (Fig. 3). How is this possible - do counts include transient birds? In Oklahoma, detection probability is included in the model, although it is not at all clear how this parameter is estimated in years without distance sampling. Please provide more details.

Validity of the findings

All the underlying data are available as R data sets. However, this means that the only way to access these data is to run the provided R script. Unfortunately, this script is not well commented, and it is difficult to get an overview of the data. Therefore:

1. I strongly suggest that the authors provide an overview of the data included in the manuscript (e.g. as a table). How many surveys were carried out, and how many birds were counted? How many nests were followed? How many birds were marked (by age class), and how many recaptures and resightings were obtained? Without this information, it is near impossible to assess the work.
2. I think the conclusions are somewhat overstated, particularly regarding the environmental covariates. In most cases, the 80% credible intervals overlap zero, which in my view indicates that there is no evidence to support the importance of the covariate in question. Some sort of assessment of the importance of the covariates would be helpful. I'm not completely up to date about how this is best done in a Bayesian context, but some sort of comparison of the performance of models with and without (i.e with constant demographic parameters) covariates might be helpful. Also, I suggest plotting some of the identified relationships before claiming e.g. that you have shown 'strong positive effects' of the Palmer drought index (l. 361).
3. You say that 'reasonable precision of demographic estimates' was achieved (l. 407). I would dispute that this is the case for survival (Fig. 5), where the 80% credible intervals in many cases almost span the entire possible range (0-1). It would be interesting to see how much the CMR data actually contribute to the likelihood - would you get similar estimates without including these data?

Additional comments

Figs. 6-8: please explain exactly what is shown in the violin plots (median, interquartile range, or whatever).

·

Basic reporting

1) The literature is well referenced and mostly relevant. However, reference to previous work also estimating demographic parameters using IPMs with limited data is missing. Most importantly, the paper titled “Use of Integrated Modeling to Enhance Estimates of Population Dynamics Obtained from Limited Data” from Schaub et al. 2007 is not cited but highly relevant. Other reference dealing with this issue: Davis et al. 2014 “An integrated modeling approach to estimating Gunnison sage-grouse population dynamics: combining index and demographic data”.

2) The raw data is supplied and I commend the authors for also providing their R script, which runs smoothly on the provided data. I think it would be very useful to provide a “readme” file of some sort that annotates/explains the raw data. Similarly, for the R script more annotations to explain the different steps and the IPM would be highly useful. For example, the formatting of the capture-recapture data (and the way it is modelled in the IPM) are not standard, so explanations would be particularly useful there. I would also suggest providing the script for the post modelling analysis (hierarchical partitioning analysis).

The paper is clearly written and well organised. Very nice and easy to read.

The figures are of exceptional quality.

Experimental design

3) I definitely think that the research fills an identified knowledge gap (a better understanding of the ecological drivers of the population dynamics of the study species. However, I do not see how the study fills a knowledge gap notably implied in the title, that “Integrated population modelling improves the utility of partially aligned data” as this is already known (see work mentioned above). As far as I can see, the proposed model has no particular feature that allows it to deal with missing/partially aligned data (all Bayesian IPMs will automatically deal with missing data by treating it as a parameter and simulating values from the likelihood). If I am wrong and the model proposed does include novelty that allows dealing with partially aligned data, then it should be better emphasised.

Otherwise, I suggest that the authors either:
i) change the focus for the biological aspect of the study, that is, assessment of ecological factors influencing the demography of the study population and mention the missing data as a feature of the data.
ii) keep the focus on the use of partially aligned data, and in this case explicitly assess the gain/loss in parameters identifiability and precision in their estimation when adding/deleting years for the different types of data, using the real dataset as a case study. Then the authors could provide more precise recommendations regarding which and how much data to collect in priority.

The research questions are otherwise well defined, relevant & meaningful.

4) The investigation performed is overall rigorous and to a high technical standard. However a goodness of fit assessment of the sub models is crucially needed (the authors should perform and provide some posterior predictive checks).

The methods are described with sufficient detail &information to replicate (but see comment above regarding a more detailed annotation of the data and script).

Validity of the findings

5) Conclusions are well stated and linked to original research question. That being said, I think that the confidence in the results seem often too strong given the high uncertainty in the estimates. Except from the effects of the PDSI and minimum temperature on clutch fate, there is always a non-negligible proportion of the posteriors that is below zero, suggesting statistically unclear effects. For example, given figure 8, I would not say that “The Palmer drought severity index had positive effects on … apparent survival”. I would recommend being more careful in the abstract, result section and discussion given this uncertainty. That being said I appreciated where the authors were careful (e.g. regarding the results on wind speed lines 388-389).

All underlying data have been provided; they seem sound as far as I can tell.

Additional comments

Line1: “improves” Although very likely, this was not explicitly investigated in the paper (compare with utility of non-integrated datasets (modelled independently)).

Lines 20-21: “we developed an IPM for partially aligned population and demographic data” it is not clear that/how this model is specifically developed for dealing with partially aligned data (not more than any other Bayesian IPM).

Line 34-37: I would specify that these effects are only clear on clutch fate.

Line 41: “Our modelling approach lays a foundation of allocating limited conservation resources…” I would rather say that it follows/confirm previous work on this (see above-mentioned references).

Line 87: perhaps use a transition word (e.g. “Additionally, “Finally”) to make it clearer that wind speed is newly introduced here.

Method section: it would be useful to provide information on sample size for the different datasets.

Line 143: was brood reduction assumed inexistent in the model? If so, is there some evidence that “partial nest mortality” is low in this species? It is a very common assumption to make and I think it is OK, it would just strengthen the validity of this assumption.

Line 188: State that demographic stochasticity was considered negligible, and that Immigration was considered absent.

Line 213: why “t-1” and not “t”? Typo?

Line 257: how is this method better/different than the more classically used LTRE contribution approach (Koons et al. 2016 and 2017)?

Line 271: “CI” Define somewhere that these are Credible Intervals and not Confidence Intervals.

Line 278: Can this decline be estimated? It is not so clear from the figures given the important temporal random variation.

Line 290: I would not call it “substantial” since the important overlap of the posterior samples with zero (Fig. 6). I would say uncertain.

Line 300: I would rather say high uncertainty (but is there any rule of thumb used to define low/moderate/high uncertainty here?). Similar comment for the description of the other effects and see general comment above regarding this issue.

Lines 325-328: I do not see how this show that CR and count should be continuous while productivity data should be segmented without further investigation (e.g. simulations). This statement is misleading.

Lines 330-331: Was it compared with non-IPM estimates? I do not see this.

Line 346: Another strategy that allows both borrowing info and flexibility would be to have random locality effects? It may be more useful with more than three populations but it could be suggested here (with existing references)?

Lines 358: what are the demographic mechanisms of the decline? It would require looking more specifically at the drops in population growth rates and see if they are mostly associated with e.g. drops in productivity or survival (e.g. using year-to-year transient LTRE contributions)?

Line 364: is wetland habitat proxi (PDSI) declining throughout the study period? It would be useful to provide a graph of the yearly variation of the three studied ecological factors. Also similarly to the comment just above, perhaps only increases (not decreases) in population growth are associated with increased PDSI?

Line 395: There are studies on small insectivorous birds that looked at the contributions of demographic rates to changes in growth rate so some could be cited rather than unpublished data.

Lines 404-407 “Furthermore, it would be ideal to jointly analyse count and demographic data to achieve comprehensive understanding and robust inference of population dynamics and underlying demographic processes (Zhao 2020).” Isn't it precisely was it done when using IPMs? I do not see what is meant here (and the relevance of the cited reference). However, I think I just do not understand.

Line 435 and 437: I would only say “IPMs” and not our IPM as it is the case for any IPM. Similarly, “out modelling approach” can be replaced with “IPMs” more generally.

Figures 4-5: it would be useful to show on these figures when the associated data collection was present for each pop.

·

Basic reporting

This is mostly OK. I do (below) comment on a number of things where I think a revision will help the paper to become better.

Experimental design

There is no experiment involved, so this is not an issue.

Validity of the findings

Seem all OK

Additional comments

Review of the PeerJ ms by Zhao et al.
(Please note that my comments are not in decreasing order of importance. )
Zhao et al. build an IPM for the snowy plower in three regions of three Southern states using a variety of available data sets that cover a total of 21 years. They study the effects on demography of various environmental covariates. Importantly, as so often, these data sets do not cover the exact same range of years and they come from different areas. Thus, they are at least in part spatially and temporally misaligned. Such misalignment occurs in virtually all instances of integrated models, where multiple data sets are combined in a model to (generally) inform on the same underlying process. Hence, they deal with questions that are important in general (i.e., for any type of integrated statistical models) as well as for one species that appears to have become a sort of conservation flagship in the US.
I liked the ms very much and don’t have any major complaints about it. Some comments that may be useful during the revision are here:
1. A key part of the story told by this ms is about misaligned data. I agree that this is an important topic in integrated models, but I think that the authors should discuss it a little bit more, since they made it one of their main topics. I suggest they highlight a little more those parts of the model where they accommodate this and add some material on this in the Discussion.
2. In the population submodel the authors fix the variance of initial population size at some small value. Please comment on why you did this? Usually, placing a very wide prior on this parameter makes the population size estimates in the initial year extremely wide, but then does not (so much) affect results in subsequent years. Was this different in your case so that you felt you had to essentially ignore this uncertainty in N[1] ?
3. Around the same place, where you use a Normal distribution throughout to represent the variability of population size rather than a discrete valued distribution such as a Poisson and binomial, can you also comment on that. Clearly, the Normal makes for better mixing of the MCMC, but it really assumes that your N's are not too close to the boundary of zero. Please comment on that, too.
4. Starting on line 403 the authors say that several recent studies have shown that count data carry information about demographic rates and then cite the Dail-Madsen developmental line of models. This is fine, as these authors formalized the estimation of demographic rates from counts, but of course this idea is much older and indeed lies at the heart of the whole business of IPMs right from the start (i.e. Besbeas et al. 2002): that the annual change in numbers of unmarked individuals can be expressed as a function of last year’s numbers and the demographic rates.

And here are some more minor comments:
- Fig. 3: can you say that the CRIs are shown as credible bands ? I would rather call them density strips, since a band to me implies constant density or color hue.
- Lines 55-56: so true. And you can perhaps mention that this data deficiency is the rule rather than the exception in demographic population assessments.
- line 66: could also cite Schaub et al. 2007 who particularly emphasize the (perhaps not so surprising) fact that data integration is particularly useful when data sets are very small.
- Line 196: might it be better to say that the counts were assumed to follow a Poisson log-normal distribution ? To clarify right away the feature of overdispersion that you allow for there.
- Line 206: that’s a strange detection function. Does it have a recognized name in the Distance sampling universe ? Why not take a more traditional choice such as a half-normal ?
- Line 264-265: nice ! Use of the full set of MCMC draws produced from a Bayesian analysis in some secondary analysis, to obtain new results with full error propagation, is done not nearly often enough.
- Line 290: yes, actually I find it rather surprising how similar the contribution of all the demographic components were estimated to be.

Marc Kéry, 6 September 2021

---

## Round 0.2 · Minor Revisions

Reviewers' comments on your work have now been received. The manuscript has been assessed by three reviewers. One reviewer indicated that the experimental design and empirical research results should be further improved. In addition, reviewers pointed out that some words in the main text were used inappropriately. I agree with this evaluation and I would, therefore, request for the manuscript to be revised accordingly.

·

Basic reporting

no comment

Experimental design

no comment

Validity of the findings

no comment

Additional comments

The authors have done an admirable job of revising the manuscript in accordance with the suggestions of the three reviewers. I have no further comments at this stage.

·

Basic reporting

No comment.

Experimental design

1) The authors have addressed my main comments. However, (regarding a previous minor comment) I still think that the authors consider immigration to be absent. I agree that the process error term would include variation in “net immigration” in cases where i) “true” (not apparent) survival is estimated (see e.g. Fay et al. 2019 Integrated population model reveals that kestrels breeding in nest boxes operate as a source population) AND ii) immigration is assumed to equal emigration overall, so that the mean net immigration is assumed to be zero.
I do not see where Schaub et al. 2013 refer to “net immigration” (immigration - emigration). They refer to “immigration” and clearly state that permanent emigration and mortality are both accounted for (and confounded) when estimating apparent survival.
In the present study, apparent (not true) survival is estimated. Therefore, emigration is already accounted for when estimating apparent survival and immigration can only be positive. If the authors wish to allow immigration to occur, then they should include it explicitly in the population model, for example as a constant positive integer (then I agree that its temporal variation can be accounted for by the process error). Otherwise, they should state that it is assumed to be absent.

2) Apologies for not asking earlier but I don’t quite understand the prior for “pcount” (gamma prior on theta). The fact that the posterior for this parameter virtually equals 1 (no uncertainty due to no chain fluctuation) seems driven by this prior choice. Why not a Uniform(0,1) on “pcount” (in the main text it is stated that Uniform priors between 0 and 1 were used for probabilities)?

Validity of the findings

3) Although I appreciate the effort, I disagree with the use of the “p-tail” statistics to describe the strength of the effects. There is a confusion here between “strong” and statistically clear effects. The strength of an effect should refer to its effect size, or its “biological significance” only. Please see the paper from Dushoff et al. (2019). “I can see clearly now: reinterpreting statistical significance” that is a nice commentary about this common misinterpretation of p values and how to address it.
If the authors want to refer to the strength of effects, they should report standardised effects sizes and state which values will be considered small, medium and large effects throughout the manuscript (e.g. for Cohen’s d, d = 0.2 is considered a 'small' effect size, 0.5 represents a 'medium' effect size and 0.8 a 'large' effect size).
Also I strongly agree with Reviewer 1 that plotting estimated relationships (point estimates and credible intervals) between demographic rates (y axis on natural scale) and the covariates of interest (x axis showing the natural range of values) would be highly helpful for the reader to evaluate (and estimate for themselves) the strength of the relationships and the uncertainty around them (such as in Zhao et al. “Land‐use change increases climatic vulnerability of migratory birds: Insights from integrated population modelling” Figure 5).

Dushoff, J., Kain, M. P., & Bolker, B. M. (2019). I can see clearly now: reinterpreting statistical significance. Methods in Ecology and Evolution, 10(6), 756-759.
Cohen, J. (1992). A power primer. Psychological bulletin, 112(1), 155.

4) Perhaps rather than saying that survival parameters were estimated with greater uncertainty in years without data (e.g. in the abstract line 31), it may be more correct to state that they could not be estimated in those years? Often, posterior chains just seem to represent the prior, with modes at 0 and 1, likely due to the fact that the prior is flat on the logit scale but then not on the natural scale (this could be changed).

5) Make sure that the file names in the R code correspond to the names of the data provided (there are underscores missing, file 3 is file 2 and file 2 is 3). The constant “narea” is never used in the model so it does not need to figure in the “jags.data” list.

6) It seems like population size for the New Mexico population cannot be estimated properly (and R hat are higher than 1.15). Is it the case for the posterior used by the authors? It may be bad luck when I ran the models, but if not it may be due to the recent changes in the code (addition of the pcount parameter).

Additional comments

Line 1 Title “reveals drivers”: add “potential”, or say instead e.g. “reveals ecological covariates of demography”. Best to avoid causal language since causality was not evidenced.

Line 301: “Indicating nearly perfect detection”. Not necessarily. This merely means that double counting is as likely as no detection (false positive and negative cancel out on average). Imperfect detection is already accounted for with the Poisson observation error. It accounts for both double counting and no detection which are assumed to be as likely on average. It may be best to not include this pcount parameter (except for the distance sampling) to avoid over-parameterisation (also see previous comment 2) regarding the prior).

Lines 308-315: “adult female (…) and adult male (…) has greater survival than juvenile female (…) and juvenile male”. Either write “on average greater survival” or provide the posterior differences and their credible intervals.

Line 352 and 355: “Impacted” “influenced”, best to change for non-causal statements (here and throughout the manuscript. Or at least use “hedging” (e.g. may influence).

Lines 355-356: “Based on these results, we recommend to prioritize data collection on population and capture-recapture surveys” it is not straightforward for me how this is based on the abovementioned results. Please clarify, or delete "based on these results".

Line 390: “drivers” ditto about causal statements.

·

Basic reporting

I had previously thought that the reporting was good and now it has gotten even better.

Experimental design

The authors have made a good job a improving the descriptions of the design of their field studies, as asked for by the two other referees. This, together with the analytical methods, are much improved now.

Validity of the findings

The findings appear all valid.

Additional comments

Here are some minor comments by line:


- line 94: 'foundation' sounds a bit grand. Why not 'causes' ?
- 119-120: but not among years ?
- 166: I must say that drought severety index is a clear misnomer ! Palmer should have multiplied his index with -1.
- 171: 'We also considered' twice in two subsequent sentences
- 213: why did you not use one of the more traditional detection functions, such as a half-normal ?
- 326: since you said this already perhaps you can precede the half-sentence in the parentheses by "recall that"
- 461: "IPMs lay ..."

---

## Round 0.3 · accepted · Accept

The authors have adequately addressed the comments raised in a previous round of review and I feel that this manuscript is now acceptable for publication.

·

Basic reporting

No comment.

Experimental design

No comment.

Validity of the findings

I am really sorry about the ongoing confusion regarding effects sizes, and I really don’t mean to slow down the publication process of this paper any further, but I am afraid I still don’t find the “standardised effect sizes” suggested as an appropriate measure of effect size. The authors divide the posterior mean effects by the width of the 80% credible interval. I am completely unfamiliar with this type of effect size (and there was no reference provided), but such effect size runs into exactly the same issue as the one I mentioned earlier for the p-tail, that is, it directly depends on the uncertainty around the estimate. For a given effect size, effects estimated with more precision will be considered stronger.
There are several possibilities for estimating effect sizes in the literature and I am not a statistician so may not know of the best options, but one option that seem reasonable and meaningful would be to divide each posterior estimate of the slopes by the associated posterior standard deviation of the (transformed) response variable, e.g. the temporal standard deviation of the logit of survival for effects modelled on survival. For example the effect size of PDSI on juvenile male survival of the Oklahoma population would be (for each posterior iteration):
Effect_size_sjf_ok <- logit_sjf_palm/sd(logit_sjf_ok[])
This should provide an estimation of the proportion of variance of each vital rate explained by each effect (and their credible interval).
But I leave it up to the authors to choose what measure they think is most appropriate and useful.

I really appreciate the addition of Figure 8, thank you!

Additional comments

No more comment, and apologies again, this is a nice and clear paper.